Geologic and anthropogenic sources of contamination in settled dust of a historic mining port city in northern Chile: health risk implications

http://orcid.org/0000-0002-1516-2622 Tapia Joseline S. 1 joseline.tapia@uach.cl
Valdés Jorge 2 3
Orrego Rodrigo 2
Tchernitchin Andrei 4 5
Dorador Cristina 6 7
Bolados Aliro 5
http://orcid.org/0000-0002-5353-1556 Harrod Chris 2 8
1 Instituto de Ciencias de la Tierra, Universidad Austral de Chile , Valdivia , Chile
2 Instituto de Ciencias Naturales Alexander von Humboldt, Universidad de Antofagasta , Antofagasta , Chile
3 Laboratorio de Sedimentología y Paleoambientes LASPAL, Universidad de Antofagasta , Antofagasta , Chile
4 Laboratorio de Endocrinología Experimental y Patología Ambiental ICBM, Facultad de Medicina, Universidad de Chile , Santiago , Chile
5 Departamento de Medio Ambiente, Colegio Médico de Chile , Santiago , Chile
6 Departamento de Biotecnología and Instituto Antofagasta, Universidad de Antofagasta , Antofagasta , Chile
7 Centre for Biotechnology and Bioengineering (CeBiB) , Antofagasta , Chile
8 Núcleo Milenio INVASAL , Concepción , Chile
Anderson Todd
Electronic publication date: 2018 Apr 24
Publication date: 2018
Volume: 6
Electronic Location ID: e4699
Received 2018 Feb 5; Accepted 2018 Apr 12
Copyright: © 2018 Tapia et al.
Copyright year: 2018
Copyright holder: Tapia et al.
License: This is an open access article distributed under the terms of the Creative Commons Attribution License, which permits unrestricted use, distribution, reproduction and adaptation in any medium and for any purpose provided that it is properly attributed. For attribution, the original author(s), title, publication source (PeerJ) and either DOI or URL of the article must be cited.
License URL: https://creativecommons.org/licenses/by/4.0/

Keywords: Copper concentrate, Mining, Polymetallic ores stockpiles, Contaminant source, City dust, Risk strategies, Hazard index, Human health

Funding: CONICYT “Programa de Inserción en la Academia” PAI-79150070 Núcleo Milenio INVASAL Chile’s government program, Iniciativa Científica Milenio from Ministerio de Economía, Fomento y Turismo J. Tapia benefited from the CONICYT “Programa de Inserción en la Academia” (PAI-79150070). C. Harrod is supported by Núcleo Milenio INVASAL funded by Chile’s government program, Iniciativa Científica Milenio from Ministerio de Economía, Fomento y Turismo. The funders had no role in study design, data collection and analysis, decision to publish, or preparation of the manuscript.

==============================
Chile is the leading producer of copper worldwide and its richest mineral deposits are found in the Antofagasta Region of northern Chile. Mining activities have significantly increased income and employment in the region; however, there has been little assessment of the resulting environmental impacts to residents. The port of Antofagasta, located 1,430 km north of Santiago, the capital of Chile, functioned as mineral stockpile until 1998 and has served as a copper concentrate stockpile since 2014. Samples were collected in 2014 and 2016 that show elevated concentrations of As, Cu, Pb, and Zn in street dust and in residents’ blood (Pb) and urine (As) samples. To interpret and analyze the spatial variability and likely sources of contamination, existent data of basement rocks and soil geochemistry in the city as well as public-domain airborne dust were studied. Additionally, a bioaccessibility assay of airborne dust was conducted and the chemical daily intake and hazard index were calculated to provide a preliminary health risk assessment in the vicinity of the port. The main conclusions indicate that the concentrations of Ba, Co, Cr, Mn, Ni, and V recorded from Antofagasta dust likely originate from intrusive, volcanic, metamorphic rocks, dikes, or soil within the city. However, the elevated concentrations of As, Cd, Cu, Mo, Pb, and Zn do not originate from these geologic outcrops, and are thus considered anthropogenic contaminants. The average concentrations of As, Cu, and Zn are possibly the highest in recorded street dust worldwide at 239, 10,821, and 11,869 mg kg−1, respectively. Furthermore, the contaminants As, Pb, and Cu exhibit the highest bioaccessibilities and preliminary health risk indices show that As and Cu contribute to elevated health risks in exposed children and adults chronically exposed to dust in Antofagasta, whereas Pb is considered harmful at any concentration. Therefore, an increased environmental awareness and greater protective measures are necessary in Antofagasta and possibly other similar mining port cities in developing countries.

Introduction

The process of economic development often results in large-scale anthropogenic impacts to the environment and inhabitants of developing countries. Previous well-reported examples of environmental pollution caused by industrial activities include: (i) the 1984 Bhopal incident in India after more than 40 tons of methyl isocyanate gas leaked from a pesticide plant (Broughton, 2005), (ii) Chernobyl, Ukraine, in 1986, after the nuclear energy plant explosion of Reactor 4 (Devell et al., 1986; Gale, 1987), (iii) the Baia Mare spill, Romania, in 2000, where nearly 100,000 m3 of cyanide and metal-rich liquid waste was released into the river system near this city (Lucas, 2001; Soldán et al., 2001), and (iv) Sukinda, India, where the most harmful form of chromium (Cr[VI]) pollutes water and inhabitants due to mining activities (Dubey, Sahoo & Nayak, 2001). As such, industrial incidents have been related to severe health issues and fatalities (Khan & Abbasi, 1999).

Despite harmful consequences that industry and uncontrolled anthropogenic activities have caused and are still contributing to (Reddy & Yarrakula, 2016), an awareness of the environmental impact of these activities is growing in many developing countries. Chile, a developing country located along the western border of southern South America, is one example where an environmental awareness has increased with time on a national scale. Among its developing characteristics is the fact that the country predominantly relies on the exploration, exploitation, and exportation of mineral resources as one of its main economic activities (De Solminihac, Gonzales & Cerda, 2017). Although often associated with environmental degradation, these activities support an elevated gross domestic product (GDP) that has allowed Chile to be included in the Organization for Economic Co-operation and Development (OECD; Ruiz-Rudolph et al., 2016).

Antofagasta Region background

Copper (Cu) is the most significant metallic resource of Chile: the country has contributed approximately 28% of the world’s Cu production since 1985 (COCHILCO, 2014). This element is naturally distributed in northern Chile, principally in the Antofagasta Region (Fig. 1A; Aroca, 2001), where large-to-giant porphyry copper deposits are found (e.g., Chuquicamata, La Escondida, and Radomiro Tomic, among others; Fig. 1B). As such, numerous large-scale mining operations and mineral deposits in the Antofagasta Region have supported the economic development of the region to the point where the per capita GDP is the highest in the country.

Figure 1 Antofagasta location.

Location maps with features, at varying scales. (A) Location of the Antofagasta Region in northern Chile; (B) Main mineral deposits and mining activities within Antofagasta Region (1, Mantos de la Luna Mine; 2, Michilla Mine; 3, Juanita Mine; 4, Mantos Blancos Mine; 5, La Negra Industrial Complex; 6, Pedro de Valdivia (ex-nitrate mine); 7, El Abra Mine; 8, Radomiro Tomic, Chuquicamata, and Ministro Hales Mines; 9, Faride, Spence, Sierra Gorda Mines, and Aconcagua treatment plant; 10, Esperanza Mine; 11, Zaldívar and La Escondida Mines; 12, Francke Mine), and Antofagasta city (black square); (C) Downtown Antofagasta. 1, Port gate; 2, Clínica Antofagasta (health institution); 3, Parque Brasil (Children’s playground); 4, Commercial center; 5 and 6, Schools (Liceo de Hombres and Liceo de Niñas, respectively); 7, Housing complex; 8, City Mall; 9, Municipal square; 10, Hospital (health institution); 11, Fruit and vegetable market; 12, Supermarket. The red circle represents an area with a 1 km distance (radius) from the port gate.

Following the War of the Pacific (1879–1884), Chile and Bolivia signed a trade agreement in 1904 which allowed Bolivian products to be exported from the Train Station and Port of Antofagasta, where resultantly, unprotected stockpiles containing metals (mainly Pb) became common at those facilities (Sepúlveda, Vega & Delgado, 2000).

It is also important to consider other natural and anthropogenic sources of contamination in Antofagasta that accompanied heightened mining and a related increase in population (by a factor of nearly 6.5 between 1895 (13,530 inhabitants; Censo, 1895) and 1960 (87,860 inhabitants; Censo, 1960)): (i) of the sparse fresh water sources in the region, the Toconce and Holajar rivers were naturally enriched in As, resulting in chronic contamination of drinking water until 1970 (Marshall et al., 2007), and (ii) in the late 1970s, La Negra Industrial Complex was established 33 km southeast of Antofagasta which included cement production (Industria Nacional de Cemento SA, Antofagasta, Chile and Inacesa, Antofagasta, Chile), lithium (Li) processing (Sociedad Chilena del Litio, Antofagasta, Chile and SQM, Antofagasta, Chile), and a smelter (Refimet, currently Altonorte of Xstrata Copper; Minería Chilena, 2013).

At the end of 1980s, elevated Pb levels in blood were reported in children attending schools and living close to the Antofagasta Port and Antofagasta Train Station facilities (Sepúlveda, Vega & Delgado, 2000; Tchernitchin et al., 2006). As a result, in the early 1990s, the stockpiles of Pb-rich materials originating from Bolivia were transferred to Portezuelo, 18 km southeast of downtown Antofagasta, in close proximity to La Negra Industrial Complex. Protective measures related to the transport and deposition of concentrate from Portezuelo to Antofagasta Port were not known to be established prior to 2010 (Chilean Ministry of Transport and Telecommunications (MTT), 2015).

Combined lung and bladder cancer mortality rates in the Antofagasta Region were the highest reported for the whole country between 1992 and 1994 (153 and 50 per 100,000 men and women, respectively), attributable to the chronic ingestion of As between 1958 and 1970 (Marshall et al., 2007). In 2014, despite documented adverse health impacts to the population of Antofagasta and relocation of the Bolivian stockpiles, the development of Cu, Mo, and Ag exploitation in the Sierra Gorda district, 140 km northeast of the city of Antofagasta (Fig. 1B; Minería Chilena, 2014), led to the construction of a holding facility inside the Antofagasta Port called the galpón (warehouse), located close to the center of the city, which currently stores Chilean Cu concentrate. Aside from Cu (nearly 32%), the concentrate contains other elements such as S, Fe, Zn, As, Pb, Mo, Sb, and Cd (Table 1 in Fuentes, Viñals & Herreros, 2009). As port operations continued, local inhabitants noticed the increase of dust on buildings etc., which led to criticism of the operations associated with the galpón and the port. As a result, social organizations were established, such as Este polvo te mata (“This dust kills you”), leading to ongoing environmental conflicts within the city.

Table 1 Sampling sites.

Sample	Location	Year	Institution	UTM E	UTM N	Elevation (m.a.s.l)	AP (km)	
1	Grecia/Salvador Reyes	2014	ISP	356925	7382802	10	0.62	
2	Grecia/Salvador Reyes	2014	ISP	356925	7382802	10	0.62	
3	Grecia 1816/21 Mayo	2014	ISP	356973	7383725	8	0.31	
4	Grecia/21 Mayo	2014	ISP	357075	7383834	11	0.45	
5	Uribe/Balmaceda Pabellón 1	2014	ISP	357008	7384065	9	0.65	
6	Uribe/Balmaceda Pabellón 1	2014	ISP	357008	7384065	9	0.65	
7	Uribe/Pabellón 2	2014	ISP	357025	7384049	10	0.64	
8	Uribe/Pabellón 2	2014	ISP	357025	7384049	10	0.64	
9	Uribe/Pabellón 3	2014	ISP	357049	7384024	10	0.62	
10	Uribe/Pabellón 3	2014	ISP	357049	7384024	10	0.62	
11	Uribe/Washington Pabellón 4	2014	ISP	357062	7384012	10	0.61	
12	Uribe/Washington Pabellón 4	2014	ISP	357062	7384012	10	0.61	
13	MOP building in front of the Port	2014	ISP	356969	7383667	7	0.26	
14	MOP building inside in front of the Port	2014	ISP	356988	7383637	8	0.24	
15	Liceo Técnico in front of the Port	2014	ISP	357027	7383797	11	0.40	
16	Colegio Técnico, frontis	2014	ISP	356996	7383771	10	0.37	
17	Edificio Colectivo Argentina	2014	ISP	357023	7383850	10	0.45	
21	Edificio Colectivo Perú 1er piso	2014	ISP	357027	7383967	10	0.56	
22	Edificio Colectivo Perú 2o piso	2014	ISP	357017	7384003	10	0.59	
23	14 de Febrero/Edmundo Pérez Zujovic	2014	ISP	357560	7386404	12	3.10	
24	Av. Edmundo Pérez Zujovic 7344	2014	ISP	357968	7389089	14	5.77	
25	Av. Edmundo Pérez Zujovic 8126-9114	2014	ISP	357562	7389860	13	6.47	
26	Colegio San Agustín	2014	ISP	357568	7390407	8	7.02	
27	Condominio Jardines del Norte VI	2014	ISP	357514	7392555	18	9.15	
AF-1	Jardín Infantil Semillita	2016	CM	357072	7383698	9	0.33	
AF-2	Jardín Infantil Semillita	2016	CM	357072	7383698	9	0.33	
AF-3	Ex Liceo Técnico A14	2016	CM	356986	7383759	9	0.35	
AF-4	Lice Técnico A14	2016	CM	357071	7383760	10	0.38	
AF-5	Lice Técnico A14	2016	CM	357071	7383790	11	0.41	
AF-6	Liceo Marta Narea A1	2016	CM	357128	7383791	12	0.44	
AF-7	Liceo Marta Narea A1	2016	CM	357156	7383822	12	0.48	
Notes:

Summary of street dust sampling locations: name of streets, sampling years, sampling institutions, UTM coordinates (WGS-84), and elevations (m.a.s.l.). AP, distance to the Antofagasta Port in kilometers (km).

Currently, the city of Antofagasta has 361,873 inhabitants (Instituto Nacional de Estadísticas (INE), 2017), which is nearly four times greater than the 1960 population (Censo, 1960). The city has grown and developed around the port (Fig. 1C); as a result, 5.7% of the national state load and 17.3% of all regional transport passes through the port (Chilean Ministry of Transport and Telecommunications (MTT), 2015). Important locations situated near the Antofagasta Port include the downtown area of Antofagasta, hospitals, schools, preschools, the municipal square, the city shopping mall, and commercial centers (Fig. 1C).

Previous sampling campaigns

Due to increased social pressure associated with the Antofagasta dust, in 2014, the Chilean State sampled city dust and human blood and urine from locations in close proximity to, and up to 9 km distance from the Antofagasta Port (Table 1). Contaminant concentrations were quantified in the <63 μm size fraction by the Chilean Public Health Institute (Instituto de Salud Pública (ISP), 2014). Elevated levels of Pb in blood and of As in urine were found in children attending preschools close to the Antofagasta Port, and elevated concentrations of a number of elements were also reported in street dust, including As, Cu, Pb, and Zn (Vergara, 2015). More recently, the Chilean College of Physicians (CM-Colegio Médico) undertook another sampling campaign in 2016 to compare concentrations with the values from 2014 for determining which contaminants were still present in the <63 μm size fraction of dust (Tchernitchin & Bolados, 2016). Details regarding the methodologies used in these previous studies are found in Table 2.

Table 2 Summary of analytical methodologies utilized in previous studies.

	Detection limit (mg · kg−1)	Instrument	Laboratory	
As	Ba	Cd	Co	Cr	Cu	Mn	Mo	Ni	Pb	V	Zn			
Antofagasta Dust	
ISP (2014)	1.0	1.0	1.0	1.0	1.0	1.0	1.0	1.0	1.0	1.0	1.0	1.0	ICP	ISP, Chile	
Tchernitchin & Bolados (2016)	1.7	2.1	1.0	0.8	1.8	2.1	2.5	0.9	2.2	2.1	0.7	3.2	ICP-OES	CENMA, Chile	
Rocks of Antofagasta	
Lucassen & Franz (1994)	N/A	Major elements XRF and trace element ICP	TU-Berlin	
Oliveros et al. (2007)	1.0	3.0	0.4	0.2	4.0	0.2	3.9	0.5	4.0	1.2	1.5	8.0	ICP-AES and ICP-MS	CRPG, France	
Rogers & Hawkesworth (1989)	N/A	Major elements ED XRF and trace elements WD XRF	Open University and Nottingham University	
Soil/sediments	
CENMA (2014)	14.6	8.9	1.8	2.4	4.6	1.0	8.5	8.3	1.4	3.7	1.5	2.7	ICP-OES	CENMA, Chile	
De Gregori et al. (2003)	N/A	FAAS (Cu), HG-AFS (As)	N/A	
Notes:

Antofagasta dust, ISP (2014) and Tchernitchin & Bolados (2016); rocks, Lucassen & Franz (1994), Oliveros et al. (2007), and Rogers & Hawkesworth (1989); soil and sediments, CENMA (2014) and De Gregori et al. (2003); N/A, information not provided.

Study objectives

Despite the recent 2014 and 2016 sampling events, no analyses were conducted to evaluate and understand the spatial distribution, variation, and sources associated with health risks by exposure to these metallic and metalloid (As) contaminants apart from their bioaccessibility. Also, regardless of well-reported improvements in the technology used to transport and store material inside the Antofagasta Port (Chilean Ministry of Transport and Telecommunications (MTT), 2015), currently there is little evidence that the environmental issues and negative health impacts on Antofagasta residents have improved, and regulations have not been put into effect to control the contamination. Therefore, considering the need to better understand and establish (i) the spatial-temporal variability of the present contaminants, (ii) the likely source of contamination through the comparison of natural background concentrations, (iii) human health risk standards associated with exposure to the dust from the Antofagasta Port, and (iv) the bioaccessibility of the contaminant elements, a health risk assessment and bioaccessibility assay were conducted, and raw data collected in 2014 (ISP, 2014) and 2016 (Tchernitchin & Bolados, 2016) was further interpreted and analyzed to improve the understanding of pollution in Antofagasta and better inform regulators and interested parties in the support of new environmental policies and regulations.

Methodology

Data compilation

Information related to the elemental concentrations of metals was compiled from previous studies of (i) street dust and (ii) the composition of geologic outcrops, weathered products, and soil in the area of Antofagasta.

Antofagasta street dust

Starting 1993, leaded fuels were banned by law in Santiago, Chile (Faiz, Weaver & Walsh, 1996), as well as in other regions of the country thereafter. In medium- to low-size coastal cities of Chile, such as Tocopilla, located at 175 km north of Antofagasta, it has been shown that vehicle emissions do not significantly contribute to ambient particulate matter (PM) concentrations in the city (Jorquera, 2009). Therefore, street dust in urban areas of Antofagasta is considered to be an indicator of metal and metalloid contamination from atmospheric deposition, and public dust samples from a number of key city locations were compiled from ISP (2014) and CM (Tchernitchin & Bolados, 2016) (Tables 1 and 2). These sampling data are simply reported from those sources and reinterpreted in the context of other factors such as the background geology, which depending on the host rock mineral composition, can weather and contribute to increased concentrations of the same contaminant elements.

Geologic outcrops, weathered products, and soil

The main geologic formations of Antofagasta correspond to Jurassic volcanic basic to intermediate rocks of the La Negra Formation and Neogene marine sedimentary rocks of La Portada Formation, constituted by sandstones and coquinas, of which the latter is a form of limestone represented by an agglomerate of shells (National Service of Geology and Mining Service (SERNAGEOMIN), 2003). Concentrations of As, Ba, Cd, Co, Cr, Cu, Fe, Mn, Mo, Ni, Pb, V, and Zn of volcanic, intrusive, metamorphic rocks, and dikes of Antofagasta were compiled from Lucassen & Franz (1994), Oliveros et al. (2007), and Rogers & Hawkesworth (1989) (Fig. 2A). In addition, unpublished data (As and Mo) from Oliveros et al. (2007) was utilized. Element concentrations from sedimentary rocks of the La Portada Formation were not available, however the composition of soil, an exogenous matrix associated with the weathering of parent rocks, was obtained from De Gregori et al. (2003; two soils around Antofagasta) and CENMA (2014; 15 background samples and two contaminated samples from Antofagasta). A brief summary of the methodologies used in these studies is presented in Table 2.

Figure 2 Antofagasta geology.

Locations with geochemical data. (A) Regional volcanic, intrusive, metamorphic rocks, dikes, and soils found in the vicinity of the city of Antofagasta; (B) Local geology and dust samples from the city of Antofagasta. The red circle represents an area with a 1 km distance (radius) from the port.

Data analysis

Spatial distribution of the geologic outcrops and dust samples

The spatial distribution of regional volcanic, intrusive, metamorphic rocks, dikes, and soils was plotted using the Geographic Information System QGIS (2.6.1 Brighton) software (Fig. 2A). Local-scale surficial geology (1:50,000), provided by the program Geología para el Ordenamiento Territorial de Antofagasta of the National Service of Geology and Mining (Servicio Nacional de Geología y Minería National Service of Geology and Mining Service (SERNAGEOMIN), 2014), was used to plot the distribution of Antofagasta dust samples and their corresponding relationship to the geologic outcrops (Fig. 2B).

Metal and metalloid statistics

To determine basic characteristics of the As, Ba, Cd, Cr, Cu, Mn, Mo, Ni, Pb, V, and Zn data, univariate statistics (e.g., mean, standard deviation, median, minimum, maximum, lower, and upper limit of confidence interval (95%)) were calculated in SYSTAT (Systat Software, San Jose, CA, USA). Principal component analysis (PCA) was conducted on loge(x + 1)-transformed, normalized data in PRIMER 7 (Clarke & Gorley, 2006). In addition, comparisons to the upper continental crust (UCC; from Rudnick & Gao, 2003) are also presented.

Geo-accumulation index and enrichment factor

To infer contaminant elements and sources of street dust, numerous studies have utilized the geoaccumulation index (Igeo; Eq. (1); e.g., Lu et al., 2009; Li et al., 2013) and enrichment factor (EF; Eq. (2); e.g., Zoller, Gladney & Duce, 1974; Lu et al., 2009), respectively.

(1) Igeo=log2(Cn1.5×Bn)

In Eq. (1), Cn corresponds to the measured concentration of metal n in the sediment and Bn is the local background value of the metal n. A factor of 1.5 was used for possible variations of the local background due to variable lithologies (Muller, 1979; Nowrouzi & Pourkhabbaz, 2014). The local background value (Bn) of each metal was obtained from volcanic, intrusive, metamorphic rocks, dikes, and soils in the region. Specific values of the Igeo index indicate the following: Igeo ≤ 0, the sample is not contaminated;

0 < Igeo ≤ 1, the sample is non- to slightly contaminated;

1 < Igeo ≤ 2, the sample is moderately contaminated;

2 < Igeo ≤ 3, the sample is moderately to highly contaminated;

3 < Igeo ≤ 4, the sample is highly contaminated;

4 < Igeo ≤ 5, the sample is highly to extremely contaminated;

Igeo > 5, the sample is extremely contaminated.

Enrichment factors (Eq. (2)) were calculated using the geochemical composition of basement rocks and soil of Antofagasta, with Fe as the normalizing element.

(2) EF=M1Eref1/M2Eref2

In Eq. (2), EF is equivalent to the Enrichment Factor (Zoller, Gladney & Duce, 1974). M1 is the metal or metalloid concentration in the sample, Eref1 is the reference element in the sample (Fe), M2 is the background concentration of the metal or metalloid, and Eref2 is the background concentration of the reference element (Fe in intrusive, volcanic, metamorphic rocks, dikes, and soil of Antofagasta). Fe, Al, or Ti have been used as reference elements in previous studies (Rule, 1986; Ergin et al., 1991; Tapia et al., 2012); however, Fe was chosen as the reference element here because it was the only conservative element present in all of the utilized data sets.

Health risk assessment

To obtain a preliminary health risk assessment associated with exposure to Antofagasta dust, the chemical daily intake (CDI; Eq. (3)) of the studied elements was calculated by ingestion exposure while considering the start of the galpón and chronic exposure as two and 70 years, respectively. Dust consumption and body weight were obtained from the United States Environmental Protection Agency (1989a).

(3) CDIingestion=C×IngR×Efreq×EdurBW×AT

In Eq. (3), CDIingestion corresponds to the chemical daily intake (in mg · kg−1 · day−1), while C is the 95% confidence upper limit of the metal or metalloid concentration (in mg · kg−1), IngR is the dust ingestion rate (200 mg · day−1 for children under six and 100 mg · day−1 for adults), Efreq is the exposure frequency (days · year−1), Edur is the exposure duration (years), BW is the average body weight, and AT is the average time of exposure (ED × 365 days for non-chronic exposure; 70 × 365 days for chronic exposure). As suggested by the United States Environmental Protection Agency (1989a) the hazard index by oral ingestion (HIingestion; Eq. (4)) of the Antofagasta dust was preliminarily obtained using the CDIingestion and reference doses (RfD) from literature (As: United States Environmental Protection Agency (1991a); Ba: Dallas & Williams (2001); Cd: United States Environmental Protection Agency (1989b); Co: Finley et al. (2012); Cr(VI): United States Environmental Protection Agency (1998); Mo: United States Environmental Protection Agency (1992); Ni (soluble salts of Ni): United States Environmental Protection Agency (1991a); V: Risk Assessment Information System (RAIS) (1991); Zn: United States Environmental Protection Agency (2005)). For Cu, the HIingestion was obtained with the minimal risk level (MRL; ATSDR, 2004), and for Pb, the HIingestion was not calculated because the RfDPb (Pb reference dose) does not exist given that Pb is considered toxic at any concentration (United States Environmental Protection Agency, 2004).

(4) HIingestion=CDIingestionRfD

In Eq. (4), HIingestion is the hazard index (by ingestion), CDIingestion is the chemical daily intake (by ingestion), and RfD is the reference dose (United States Environmental Protection Agency, 1989a). The greater the value of HI above unity (1), the greater the level of concern. Therefore HI ≤ 1 suggests unlikely adverse health effects whereas HI > 1 suggests the probability of adverse health effects (Luo et al., 2012; Massey, Kulshrestha & Taneja, 2013). For As, in the case of HI > 1, health effects have been related to cellular necrosis and cancer (United States Environmental Protection Agency, 1991b).

Bioaccessibility assay

Health effects caused by exposure to the different components of dust mainly occur by ingestion of small-sized PM. Fine and ultrafine particles (smaller than 2.5 μm in diameter; ≤PM2.5) present a health risk due to their entrance into the bloodstream from lung alveoli, whereas larger particles (>PM2.5) are momentarily retained in the bronchi and bronchiole and are then expelled from the cilia of epithelial cells in the respiratory tract. Subsequently, these particles are swallowed, and once in the stomach, gastric hydrochloric acid at pH 2.0 partially solubilize components which constitute the bioaccessible fraction of the dust used to estimate the amount available for absorption across the gastrointestinal barrier (Bradham et al., 2017). Therefore, a higher bioaccessibility of a contaminant element signifies that a larger fraction will enter the human bloodstream.

To compliment the health risk assessment, a bioaccessibility assay was performed at three sites close to the Antofagasta Port. Following existent methodologies (Cortés et al., 2015), this assay was performed in the dust fraction of a diameter less than 63 μm (mesh #230). Dust was digested with chloridric acid at a pH of 2 and a temperature of 37 °C, for 2 hours, in order to resemble the digestive conditions of the human stomach. Three replicates were used for every sample and the standard recovery varied between 82% and 98%. Elements within the digested residue were quantified by ICP-OES.

Results

Statistical summary and spatial variation

Compared to the UCC mean concentrations (Rudnick & Gao, 2003; Table 3), Antofagasta dust concentrations of Co, Cr, Mn, and V from 2014 and 2016 were similar, whereas Ba and Ni were lower. These elements also exhibited a lower relative standard deviation (σ ÷ mean × 100) when compared to the mean concentration of all considered data. This variability ranged from 21% to 69% for V and Ba, respectively (Table 3). Comparing concentrations from 2014 and 2016 at locations within a 0.5 km distance from the Antofagasta Port, mean values of Ba (+96%), Co (+71%), Ni (+33%), and V (+32%) increased, and Cr (−3%) and Co (−3%) showed a slight decrease.

Table 3 Basic statistical summary of Antofagasta dust.

	Distance to AP (km)	n	Mean (mg · kg−1)	σ (mg · kg−1)	Relative σ %	Median (mg · kg−1)	Range (mg · kg−1)	95% confidence limits (mg · kg−1)	Correlation to distance to AP	2016–2014 (mg · kg−1)	UCC (mg · kg−1)	Element/UCC	
As	<0.5	14	376	347	92	272	127	1492	175	577	−0.46	−186	4.8	78	
	0.5–1	12	148	99	67	117	42	350	81	194	−0.63	–		31	
	>1	5	77	14	18	72	68	101	40	94	0.18	–		16	
	<0.5 to >1	31	239	269	112	175	42	1492	141	338	−0.28	–		50	
Ba	<0.5	14	216	136	63	234	27	418	137	294	0.33	207	624	0.3	
	0.5–1	12	112	71	63	113	30	264	67	157	−0.71	–		0.2	
	>1	5	293	132	45	225	198	510	129	457	0.50	–		0.5	
	<0.5 to >1	31	188	129	69	181	27	510	141	236	0.39	–		0.3	
Cd	<0.5	14	81	37	46	77	22	151	59	102	−0.25	35	0.09	895	
	0.5–1	12	20	13	68	18	4	52	11	28	−0.48	–		221	
	>1	5	6	1	17	6	4	7	4	7	−0.56	–		63	
	<0.5 to >1	31	45	42	93	28	4	151	30	60	−0.43	–		500	
Co	<0.5	14	22	14	61	21	9	64	15	30	0.00	16	17.3	1.3	
	0.5–1	12	12	4	34	11	6	20	9	14	0.01	–		0.7	
	>1	5	15	1	8	15	13	15	13	16	−0.76	–		0.8	
	<0.5 to >1	31	17	11	62	15	6	64	13	21	−0.13	–		1.0	
Cr	<0.5	14	62	12	19	62	47	82	55	69	−0.68	−2	92	0.7	
	0.5–1	12	46	12	25	43	28	76	38	53	−0.04	–		0.5	
	>1	5	94	15	16	93	71	110	76	112	−0.95	–		1.0	
	<0.5 to >1	31	61	20	33	55	28	110	53	68	0.57			0.7	
Cu	<0.5	14	17914	9897	55	16114	6725	46898	12199	23628	−0.62	−3044	28	640	
	0.5–1	12	6103	5059	83	4735	1103	17047	2889	9317	−0.61	–		218	
	>1	5	2287	187	8	2188	2153	2601	2054	2519	−0.06	–		82	
	<0.5 to >1	31	10821	9816	91	7874	1103	46898	7221	14422	−0.40	–		386	
Mn	<0.5	14	540	183	34	550	20	771	435	646	−0.20	−18	774	0.7	
	0.5–1	12	475	98	21	485	280	643	412	537	0.28	–		0.6	
	>1	5	675	66	10	683	590	761	594	756	0.70	–		0.9	
	<0.5 to >1	31	537	153	28	545	20	771	481	593	0.40	–		0.7	
Mo	<0.5	14	128	70	55	147	12	227	88	169	−0.11	93	1.1	117	
	0.5–1	12	35	48	135	15	5	156	5	66	−0.74	–		32	
	>1	5	22	2	9	22	19	24	19	25	0.87	–		20	
	<0.5 to >1	31	75	74	98	31	5	227	48	102	−0.33	–		68	
Ni	<0.5	14	33	9	29	37	44	17	27	38	0.01	11	47	0.7	
	0.5–1	12	23	11	49	20	11	46	16	30	−0.50	–		0.5	
	>1	5	32	5	16	30	26	38	26	39	−0.61	–		0.7	
	<0.5 to >1	31	29	11	36	30	11	46	25	33	0.06	–		0.6	
Pb	<0.5	14	1071	1091	102	739	28	3968	441	1700	−0.71	−936	17	63	
	0.5–1	12	518	481	93	412	109	1924	212	824	−0.49	–		30	
	>1	5	164	31	19	165	125	209	125	203	−0.87	–		10	
	<0.5 to >1	31	710	852	120	486	28	3968	398	1023	−0.30	–		42	
V	<0.5	14	104	23	22	106	57	138	91	117	0.21	33	97	1.1	
	0.5–1	12	86	17	20	87	58	113	75	97	0.01	–		0.9	
	>1	5	90	8	9	93	77	97	80	100	0.86	–		0.9	
	<0.5 to >1	31	95	20	21	94	57	138	87	102	−0.07	–		1.0	
Zn	<0.5	14	20351	10378	51	18320	4792	40062	14359	26343	−0.15	11392	67	304	
	0.5–1	12	6022	3943	65	4721	2029	15868	3517	8527	0.04	–		90	
	>1	5	2155	653	30	2102	1513	3149	1344	2965	−0.46	–		32	
	<0.5 to >1	31	11869	10743	91	6543	1513	40062	7929	15810	−0.41	–		177	
Notes:

Number of data (n), mean, standard deviation (σ), relative σ (σ ÷ mean × 100), median, range, lower and upper 95% confidence interval. All values are in mg · kg−1. AP, Antofagasta port; Antofagasta dust data from CM (Tchernitchin & Bolados, 2016) and ISP (2014); UCC, upper continental crust values (Rudnick & Gao, 2003).

The elements As, Cd, Cu, Mo, Pb, and Zn showed mean concentrations that were 2 (As, Mo, and Pb) and 3 (Cd, Cu, and Zn) orders of magnitude higher than the UCC. They also displayed the highest relative standard deviations when all data were considered, ranging from 91% for Cu and Zn to 120% for Pb (Table 3). Concentrations of As (−49%), Cu (−17%), and Pb (−87%) showed a significant decrease in their mean values between 2014 and 2016, while Cd (+44%), Mo (+72%), and Zn (+56%) showed increased mean concentrations within a 0.5 km radius of the Antofagasta Port (Table 3). Despite the high observed variability in element concentrations of the Antofagasta dust, the highest concentration values of As, Cd, Cu, Mo, Pb, and Zn were evident near the facility (Fig. 3). Negative Pearson correlations between these elements and distance to the Antofagasta Port range between −0.28 and −0.43 (considering all data), indicating that the element concentrations decreased with distance from the port (Table 3). The elements Co, Ni, and V showed slightly higher values near Antofagasta Port that are different from Ba, Cr, and Mn which tended to increase as a function of distance from the port. However, all concentrations are within the same order of magnitude. For these elements, Pearson correlations are lower than 0.57 (Table 3).

Figure 3 Box plots of Antofagasta dust.

Element concentrations versus distance from the Antofagasta Port (AP). The box–whisker plots show the variation in metal concentrations in street dust samples collected at various locations in Antofagasta based on their relative distance from the Port of Antofagasta (nb: in the box–whisker plots, the center vertical line shows the median value, while the length of each box shows the range within which the central 50% of the values fall, with the box edges showing the first and third quartiles (the interquartile range). Whiskers show values that the range of observed values that fall within 1.5 the interquartile range. The y-axis shows a log10-scale which differs between individual figures). Test statistics reflect results of the Kruskal–Wallis non-parametric ANOVA. (A) barium; (B) cobalt; (C) chromium; (D) manganese; (E) nickel; (F) vanadium; (G) arsenic; (H) cadmium; (I) copper; (J) lead; (K) molybdenum; (L) zinc.

Finally, the PCA of the data identified two main associations. These were related to samples located less than 0.5 km and greater than 0.5 km away from the Antofagasta Port (Fig. 4).

Figure 4 Principal component analysis of Antofagasta dust.

Principal component analysis (PCA) ordination including vectors showing relative correlation strength between principal components (PCs) and concentrations of different metals in street dust collected in Antofagasta. The circles are filled with distinct colors to show their relative proximity to the main gate of the Port of Antofagasta.

Contaminants and sources

Results for the Igeo index (Muller, 1979) are shown in Table 4. Two groups are clearly observed: the non-contaminants, including Ba, Co, Cr, Mn, Ni, and V, and the contaminants, which include As, Cd, Cu, Mo, Pb, and Zn. With the exception of As indices calculated with a soil background, all elements classified as contaminants are considered extreme contaminants within a 0.5 km distance from the Antofagasta Port (Table 4).

Table 4 Geo-accumulation index of Antofagasta dust.

	Soil	Metamorphic rocks	Volcanic rocks	Intrusive rocks	Igneous dikes	
	<0.5 km	0.5–1 km	>1 km	<0.5 km	0.5–1 km	>1 km	<0.5 km	0.5–1 km	>1 km	<0.5 km	0.5–1 km	>1 km	<0.5 km	0.5–1 km	>1 km	
As	4	3	2	–	–	–	5	4	3	7	6	5	–	–	–	
Ba	–	–	–	0	−1	0	−1	−2	−1	−1	−2	0	−1	−2	0	
Cd	5	3	1	–	–	–	–	–	–	–	–	–	–	–	–	
Co							−1	−2	−1	−1	−2	−1	–	–	–	
Cr	1	0	1	0	−1	0	0	−1	0	−1	−2	−1	−1	−2	−1	
Cu	6	5	4	7	6	4	7	6	4	7	6	4	6	5	4	
Mn	–	–	–	−2	−2	−1	−2	−2	−2	−1	−2	−1	−1	−1	−1	
Mo	–	–	–	–	–	–	6	4	3	5	4	3	–	–	–	
Ni	0	0	0	0	−1	0	0	−1	0	0	−1	0	0	−1	0	
Pb	6	5	3	–	–	–	6	5	4	8	7	5	–	–	–	
V	–	–	–	−2	−2	−2	−2	−2	−2	−2	−2	−2	−2	−2	−2	
Zn	7	5	4	7	6	4	6	4	3	8	6	5	8	6	5	
Notes:

Geo-accumulation index (Muller, 1979). Igeo ≤ 0, non-contaminated; 0 < Igeo ≤ 1, non- to slightly contaminated; 1 < Igeo ≤ 2, moderately contaminated; 2 < Igeo ≤ 3, moderately to highly contaminated; 3 < Igeo ≤ 4, highly contaminated; 4 < Igeo ≤ 5, highly to extremely contaminated; Igeo > 5, extremely contaminated. Below the sample type, distances are given from the Antofagasta Port.

The EF results are summarized in Table 5. The same elements classified as non-contaminants exhibit low EFs (generally lower than 3), indicating that they likely originate from volcanic, intrusive, metamorphic rocks, dikes, or soils within the immediate areas of Antofagasta. Conversely, contaminant elements exhibit higher EFs near the port, which suggests that the source is not related to the geology of Antofagasta and is more concentrated close to this facility (Table 5).

Table 5 Enrichment Factors obtained in Antofagasta dust.

	Soil	Metamorphic rocks	Volcanic rocks	Intrusive rocks	Igneous dikes	
	<0.5 km	0.5–1 km	>1 km	<0.5 km	0.5–1 km	>1 km	<0.5 km	0.5–1 km	>1 km	<0.5 km	0.5–1 km	>1 km	<0.5 km	0.5–1 km	>1 km	
As	7	4	2	–	–	–	91	50	22	235	130	55	–	–	–	
Ba	–	–	–	2	1	3	1	1	2	1	1	2	3	2	4	
Cd	20	7	2	–	–	–	–	–	–	–	–	–	–	–	–	
Co	–	–	–	–	–	–	1	1	1	1	1	1	–	–	–	
Cr	1	1	1	2	2	3	2	2	3	1	1	1	2	2	3	
Cu	45	21	7	313	150	46	340	163	50	236	113	35	364	174	53	
Mn	–	–	–	1	1	1	1	1	1	1	1	1	2	2	2	
Mo	–	–	–	–	–	–	135	52	26	70	27	14				
Ni	1	1	1	2	2	2	2	2	2	2	2	2	4	4	4	
Pb	35	24	6	–	–	–	196	133	34	365	248	64	–	–	–	
V	–	–	–	1	1	1	1	1	1	1	1	1	1	2	1	
Zn	74	31	9	365	151	44	172	71	21	476	198	58	934	388	113	
Notes:

Enrichment Factors obtained in Antofagasta from volcanic, intrusive, and metamorphic rocks as well as from dikes and soils. Values close to 1 indicate that the material originates from the same parent material (Zoller, Gladney & Duce, 1974). High values signify that they do not originate from that parent material. Below the sample type, distances are given from the Antofagasta Port.

Health risk assessment and bioaccessibility of dust

The CDIs and HIs related to Antofagasta dust ingestion are summarized in Table 6. As previously defined, if a child between zero and six years of age ingests 200 mg · day−1 of Antofagasta dust, on average, during two years of exposure (or from the start of operation of the galpón), the HIs related to As and Cu are elevated at all sampling points, even 3 km away from the Antofagasta Port. Conversely, Cd and Zn represent a low hazard 1 and 0.5 km from the port, respectively (Table 6B). For adults living in Antofagasta with a mean body mass of 70 kg that consume 100 mg · day−1 of dust, on average, over 70 years (signifying chronic exposure), As and Cu are moderately hazardous elements within 0.5 km of the port and slightly hazardous within 1 km of the facility (Table 6B). With the exception of Pb, which does not have a RfD value, given that it is highly hazardous at any concentration (United States Environmental Protection Agency, 2004), the elements Ba, Co, Cr, Mn, Mo, Ni, and V are not considered harmful to infants and adults.

Table 6 Chemical daily intake and Hazard Index of Antofagasta dust.

(A) CDI: chemical daily intake (ingestion)	
	95% confidence upper limit (mg · kg−1)	Two year exposure	Lifetime exposure	
				CDI	CDI	CDI	CDI	CDI	CDI	
			(mg · kg−1day−1)	(mg · kg−1day−1)	
	<0.5 km	0.5–1 km	>1 km	<0.5 km	0.5–1 km	>1 km	<0.5 km	0.5–1 km	>1 km	
As	577	194	94	0.007	0.002	0.001	0.001	0.000	0.000	
Ba	294	157	457	0.004	0.002	0.006	0.000	0.000	0.001	
Cd	102	28	7	0.001	0.000	0.000	0.000	0.000	0.000	
Co	30	14	16	0.000	0.000	0.000	0.000	0.000	0.000	
Cr	69	53	112	0.001	0.001	0.001	0.000	0.000	0.000	
Cu	23628	9317	2519	0.295	0.116	0.031	0.034	0.013	0.004	
Mn	646	537	756	0.008	0.007	0.009	0.001	0.001	0.001	
Mo	169	66	25	0.002	0.001	0.000	0.000	0.000	0.000	
Ni	38	30	39	0.000	0.000	0.000	0.000	0.000	0.000	
Pb	1700	824	203	0.021	0.010	0.003	0.002	0.001	0.000	
V	117	97	100	0.001	0.001	0.001	0.000	0.000	0.000	
Zn	26343	8527	2965	0.329	0.107	0.037	0.038	0.012	0.004	
(B) HI: hazard index (ingestion)	
		RfD (mg · kg−1day−1)	Two year exposure	Lifetime exposure	
				HI	HI	HI	HI	HI	HI	
			<0.5 km	0.5–1 km	>1 km	<0.5 km	0.5–1 km	>1 km	
As	RfD	0.0003	24	9	4	3	1	0	
Ba	RfD	0.0700	0	0	0	0	0	0	
Cd	RfD	0.0010	1	1	0	0	0	0	
Co	RfD	0.0300	0	0	0	0	0	0	
Cr	RfD [Cr(VI)]	0.0030	0	0	0	0	0	0	
Cu	MRL	0.0100	30	12	3	3	1	0	
Mn	RfD	0.1400	0	0	0	0	0	0	
Mo	RfD	0.0050	0	0	0	0	0	0	
Ni	RfD	0.0200	0	0	0	0	0	0	
Pb	–	–	–	–	–	–	–	–	
V	RfD	0.0070	0	0	0	0	0	0	
Zn	RfD	0.3000	1	0	0	0	0	0	
Notes:

Chemical daily intake and Hazard Index. Values above 1 indicate health issue risks (Luo et al., 2012). RfD: reference dose (references of RfD values are in the text).

Results of the mean bioaccessibility of contaminant elements at three sites close to the port are presented in Table 7. It is apparent that As has the highest bioaccessibility (50%), signifying that it can more readily enter the human bloodstream in comparison to the other contaminant elements. The highest bioaccessibility of As is followed by Pb (26%), Cu (20%), Zn (16%), and Cd (10%), whereas Mo does not present a heightened potential of bioaccessibility (Table 7).

Table 7 Bioaccessibility of Antofagasta dust.

		AFA-237	AFA-238	AFA-239	Mean bioaccessibility (%)	Detection limit (mg · kg−1)	Quantification limit (mg · kg−1)	Reference material recovery (%)	
		Port Gate	Window in front of the galpón	Building	
Coordinates	UTM E	356907	356899	357208	
UTM N	7383395	7383305	7384279	
As	Total (mg · kg−1)	243	92	85	50	1.7	5.5	81.6	
Bioaccessible (mg · kg−1)	27	58	65	
% of bioaccessibility	11	63	76	
Cd	Total (mg · kg−1)	62	34	17	10	1.0	3.4	83.2	
Bioaccessible (mg · kg−1)	2	2	3	
% of bioaccessibility	3	6	21	
Cu	Total (mg · kg−1)	15246	4157	3737	20	2.1	6.9	95.4	
Bioaccessible (mg · kg−1)	78	1024	1342	
% of bioaccessibility	1	25	36	
Mo	Total (mg · kg−1)	197	31	14	0	0.9	3.0	87.8	
Bioaccessible (mg · kg−1)	0	0	0	
% of bioaccessibility	0	0	0	
Pb	Total (mg · kg−1)	666	371	335	26	2.1	7.0	89.4	
Bioaccessible (mg · kg−1)	4	127	146	
% of bioaccessibility	1	34	44	
Zn	Total (mg · kg−1)	19692	8821	3021	16	3.2	10.6	98.1	
Bioaccessible (mg · kg−1)	49	941	1135	
% of bioaccessibility	0	11	38	
Note:

Bioaccessibility assay of three dust samples located in close proximity to the Antofagasta Port.

Discussions

Comparison to worldwide city dust

The results of this analysis of existing street dust data from Antofagasta indicate that As, Cd, Cu, Mo, Pb, and Zn are contaminants and their notably high concentrations are not related to the geochemistry of outcrops or soil of the city. To compare these results in a national and international context, the mean and standard deviation of these contaminants and the non-contaminant elements (Ba, Co, Cr, Mn, Ni, and V) were compared to: (i) dust accumulated in Platanus orientalis leaves of the main east–west highway of Santiago (Alameda), the capital of Chile (13 sites, Tapia et al., 2009), (ii) resuspended dust from Fushun, China (17 dust samples collected from unfrequently cleaned windowsills or plat floors), a coal-based city (Kong et al., 2011), (iii) street dust from Baoji, an important industrial city in Northwest China which has experienced a rapid urbanization and industrialization during the last decades (Lu et al., 2009), (iv) street dust from Zhuzhou, a heavily industrialized city in central China (Li et al., 2013), (v) dust from the industrial area of Qingshan district (QS) in Wuhan, China, one of the largest metropolises in that country (Zhu et al., 2013), and (vi) road dust from the Islamabad Expressway (Faiz et al., 2009), one of the busiest roads in the capital of Pakistan (Table 8).

Table 8 City dust from Antofagasta.

	Chile	China	Pakistan	
Antofagasta (mg · kg−1)	Santiago (mg · kg−1)	Fushun (mg · kg−1)	Baoji (mg · kg−1)	Zhuzhou (mg · kg−1)	QS (mg · kg−1)	Islamabad (mg · kg−1)	
As	239 ± 269	12 ± 1.9	–	–	89 ± 183	32 ± 20	–	
Ba	188 ± 129	411 ± 124	–	–	–	1,610 ± 984	–	
Cd	45 ± 42	0.8 ± 0.19	–	–	41 ± 117	2.8 ± 1.8	5.0 ± 1.0	
Co	17 ± 11	11 ± 2.3	139 ± 179	–	13 ± 11	20 ± 12	–	
Cr	61 ± 20	38 ± 11	5,334 ± 10,667	–	125 ± 54	172 ± 96	–	
Cu	10,821 ± 9,816	669 ± 567	149 ± 177	123 ± 43	139 ± 148	213 ± 180	52 ± 18	
Mn	573 ± 153	619 ± 109	–	–	–	–	–	
Mo	75 ± 74	73 ± 27	–	–	6.4 ± 12.4	7.2 ± 3.9	–	
Ni	29 ± 11	24 ± 6.4	302 ± 555	49 ± 30	40 ± 16	38 ± 14	23 ± 6	
Pb	710 ± 852	127 ± 50	–	408 ± 296	956 ± 2,815	336 ± 191	104 ± 29	
V	95 ± 20	54 ± 11	14.6 ± 4.1	–	–	–	–	
Zn	11,869 ± 10,743	943 ± 411	–	715 ± 320	2,379 ± 5,145	1,250 ± 889	116 ± 35	
Notes:

City dust from Antofagasta (ISP, 2014; Tchernitchin & Bolados, 2016) and Santiago (Tapia et al., 2009), Chile; Fushun (Kong et al., 2011), Baoji (Lu et al., 2009), Zhuzhou (Li et al., 2013), and the Qingshan district (QS) in Wuhan (Zhu et al., 2013), China; and the Islamabad Expressway in Pakistan (Faiz et al., 2009).

The concentrations of the contaminants As, Cu, and Zn in Antofagasta dust are strikingly high in comparison to street or urban dusts from heavily industrialized cities of Chile, China, or Pakistan. Mn and Mo concentrations are similar to Santiago city dust, while Pb and Cd concentration values are lower and comparable, respectively, to Zhouzhou dust. On the contrary, resuspended dust from the coal-based city of Fushun contains the highest values of Co, Cr, and Ni, and dust from the industrial QS district presents the highest concentrations of Ba (Table 8).

Anthropogenic sources of contamination

Based on the results of this study, there exists a strong correlation in Antofagasta between elevated concentrations of several key contaminants and their proximity to the Antofagasta Port. This indicates that materials stored in the port represent a source of As, Cd, Cu, Mo, Pb, and Zn that is measurable in Antofagasta city dust. In Table 4, these contaminants show higher Igeo values near the port; therefore, contamination increases close to the facility. In addition, their EFs were higher near the port (Table 5). This relationship is not shown by the non-contaminant elements (Tables 4 and 5), as some of their EFs are equal to 1, indicating that Ba might originate from volcanics, Co from volcanics or intrusives, Cr from intrusives or soil, Mn from metamorphic, volcanic, or intrusive rocks, Ni from soil, and V from all rocks of the city (Table 5). Conversely, EFs obtained for As, Cd, Cu, Mo, Pb, and Zn were generally greater than 20, indicating that their sources cannot be from rocks and soil present in the city (Table 5). To support this conclusion, other studies conducted in marine environments of the Bay of Antofagasta have showed that enrichment of metals such as Cu, Pb, and Zn in coastal waters, bottom sediments, and benthic organisms are related to industrial activities developed along the coastal border of the bay (Salamanca et al., 2000; Lépez, Furet & Aracena, 2001; Salamanca, Jara & Rodríguez, 2004; Valdés et al., 2010, 2011, 2014, 2015; Calderón & Valdés, 2012).

As potential sources of anthropogenic contamination, Chilean Cu concentrates and Bolivian stockpiles are addressed individually in the two following subsections.

Cu concentrate

Chilean Cu concentrates are notably rich in As, containing on average from 1% or 10,000 mg · kg−1 (Cantallopts, 2015) to 2.5% or 25,000 mg · kg−1 (Herreros et al., 2003) of this metalloid, indicating that the Cu concentrate stored in the Antofagasta Port represents an important source of As. Indeed, the average As in Antofagasta dust (239 mg · kg−1) represents only 2.4% of the average As concentration of Chilean Cu concentrate (10,000 mg · kg−1).

The Sierra Gorda Mine, which is situated within a Cu–Mo porphyry deposit (Brunetti, 2011), produces 120,000 tons of Cu, 50 million pounds of Mo, and 10 million pounds of Au each year (Sierra Gorda website), meaning that the Cu concentrates of this mine (and the materials stored in the galpón) are rich in Mo. This is supported by the fact that many of the most important porphyry copper deposits of Chile exploit Mo as a byproduct; for instance, there has been an average of (i) 9,212 tons every year, since 1997, of Mo in the concentrate of Chuquicamata and Radomiro Tomic and (ii) 8,572 tons, since 2000, in the concentrate of Los Pelambres (COCHILCO, 2016). Chilean porphyries can also be related to Zn anomalies (e.g., La Escondida; Garza, Titley & Pimentel, 2001). Therefore, Cu concentrate stored in the Antofagasta Port likely also acts as a source of the high concentrations of As, Mo, and Zn recorded in street dust.

Bolivian stockpiles

The elementary concentrations of stockpiles from Bolivia that were stored in the port until the 1990s are unknown, yet they contained Pb and Zn (El Mercurio, 2010). This reflects the fact that Bolivian mineral deposits are typically polymetallic. For instance, Cerro Rico from Potosí was the largest silver (Ag) deposit known and is associated with base minerals such as cassiterite (SnO2), sphalerite ((Zn,Fe)S), and galena (PbS); the San Cristóbal District contains 2.0 oz · t−1 Ag, 1.67% Zn, and 0.58% Pb with minerals such as galena, sphalerite, pyrite (FeS2), and chalcopyrite (CuFeS2); and Pulacayo hosts minerals such as sphalerite, tetrahedrite ((Cu,Fe)12Sb4S13), freibergite ((Ag,Cu,Fe)12(Sb,As)4S13), argentiferous galena, and chalcopyrite (Kamenov, Macfarlane & Riciputi, 2002). These polymetallic mineral ores are related to Pb, Zn, and Cu, which could explain the presence of these elements in dust sampled in close proximity to the Antofagasta Port. In addition, minerals containing Zn, Pb, and Cu are natural sources of Cd (ATSDR, 2012), which could explain its enrichment near the Antofagasta Port.

Health risk

The bioaccessibility of dust components was investigated in solid materials (i.e., mud and sediment) of Chañaral, Atacama Region (300 km south of Antofagasta), an environment similar to Antofagasta, revealing that the bioaccessibility percentages of some contaminant elements range from: 26% to 49% for As, 24% to 84% for Cu, 67% to 96% for Pb, and 19% to 79% for Zn (Cortés et al., 2015). In the case of the three sites studied in relation to dust of the Antofagasta Port (Table 7), As is highly bioaccessible (50%), followed by Pb (26%) and Cu (20%). These significant bioaccessibilities are supported by the fact that As and Cu show the highest HIs in the street dust (Table 6B).

The health effects of chronic exposure to toxic metals and metalloids are well-known. For example, following chronic exposure of 30 or more years to As, the probability of mortality increases dramatically due to lung and bladder cancers. In addition, prenatal or infant exposure to low levels of these elements, through the mechanism of epigenetic imprinting, can cause irreversible biochemical changes that promote the development of various organic diseases or neurobehavioral alterations in later years (Tchernitchin et al., 2013; Tchernitchin & Gaete, 2015). Also, the prenatal exposure to As increases the probability of mortality due to bronchiectasis between the ages of 30–49 years (Smith & Steinmaus, 2009); prenatal or infant exposure to Pb affects reproductive functions, decreases intelligence, and causes serious neurobehavioral changes in later stages of life (Tchernitchin et al., 2013).

Recommendations and considerations

Street dust from Antofagasta likely represents the highest recorded concentrations of As, Cu, and Zn in city dust worldwide. This result is not surprising, given that Chile is the most important producer of Cu, Chilean ores, and concentrates, acting as a plentiful source of As, Mo, and to a lesser extent Zn. Mining development in Chile continues to be predominant; for example, La Negra Industrial Complex has grown to 200 ha and 120 companies, including a Cu smelter. Previous studies, such as Ruiz-Rudolph et al. (2016), show that Chilean communes with this type of facility suffer higher mortality rates.

Although the Antofagasta Port is an important employer in the city, storing Cu concentrate and polymetallic stockpiles there for long-term periods is not appropriate due to health concerns. These materials have caused disease in children in the past (Sepúlveda, Vega & Delgado, 2000) and continue to cause elevated concentrations of pollutants in children’s blood and urine (Vergara, 2015). The results of sampling show that street dust contains concentrations of dangerous contaminants that are high enough to threaten the health of people (especially children) living and working in close proximity to Antofagasta Port. A precautionary measure previously undertaken as of 2010 for some concentrates stored in Portezuelo is that they have been and are now transported to the Antofagasta Port by hermetically sealed and cleaned trucks (Chilean Ministry of Transport and Telecommunications (MTT), 2015). Nonetheless, once stationary in Antofagasta, concentrates could be displaced by wind that moves preferentially in the SW direction; hence, special attention must be paid to the dispersion of contaminants from concentrates and stockpiles to the NE, especially during windy days.

As seen in resuspended dust from Fushun, China, some elements can be highly accumulated in dust (e.g., Cr 5,334 ± 10,667 mg · kg−1; Kong et al., 2011), especially when it is from long-term uncleaned surfaces. Antofagasta is located within the world’s driest desert, and as such, there is a very low probability for rain to remove dusts. Cleaning surfaces with wet mops twice a month and frequently washing children’s hands in locations close to the port are precautionary measures that can be taken on an individual level to prevent the ingestion of city dust, lowering Pb values in blood (Charney et al., 1983).

As a long-term goal, existing, and prospective developments located in close proximity to the port should consider the high risk of contamination and develop relevant strategies to minimize exposure to contaminated dust. Operations within the port itself should be modified (through regulations, safety guidelines, and practical measures). Additionally, it will be important for the regional and national governments to invest in suitable health care (e.g., the training of specialists and development of suitable medical facilities).

The future management and prevention of contamination in Antofagasta could also be strengthened by the implementation of a local high-quality geochemistry laboratory that allows for fast, efficient, and accurate measurements of contaminants through the utilization of specific environmental matrices where pollutant speciation can be efficiently determined. Considering that the mining industry will continue to be a profitable venture in Chile that provides jobs and income as other commodities become extensively exploited in this region (e.g., Li), the implementation and management of a laboratory of this type is necessary in this growing and highly contaminated city and region.

Finally, it is important to note that the contamination and problems described are likely not restricted to Antofagasta, or to Chile. Wherever metals are extracted, stored, and transported, there is a risk that workers and the public can be exposed to contamination and it is important that this is recognized by all parties. Chile is rapidly moving toward the status of a developed country, however environmental protection measures have not yet developed at the same rate as its economy. Industrial regulations must ensure safe levels of pollutants as is defined in the Constitution of Chile which states that Chilean citizens “have the right to live in a contamination-free environment” (Gobierno de Chile, 1980).

Conclusions

Main conclusions related to Antofagasta dust contamination are summarized below while considering the local geology, health risks, and bioaccessibility of the contaminant elements. These items are followed by potential short- and long-term recommendations and considerations that could reduce associated adverse health impacts to Antofagasta residents.

Street dust from the city of Antofagasta in northern Chile contains Ba, Co, Cr, Mn, Ni, and V that likely originate from intrusive, volcanic, metamorphic rocks, dikes, or soil of the city.

Antofagasta city dust is contaminated with As, Cd, Cu, Mo, Pb, and Zn, and this contamination likely originates from the Antofagasta Port (an anthropogenic source).

The mean concentrations of As (239 mg · kg−1), Cu (10,821 mg · kg−1), and Zn (11,869 mg · kg−1) are likely the highest in street dust worldwide.

Heightened bioaccessibility in Antofagasta dust was found for As, Cu, and Pb, indicating that comparatively high fractions could be absorbed into the human bloodstream.

The bioaccessibility and hazard indices indicate that As, Cu, and Pb in dust represent a health risk to children or adults chronically exposed to Antofagasta dust.

Potential recommendations and considerations include: Short term: Discussions with regulators and relevant parties in industry regarding the significance of the results in relation to the city’s occupants. Prospective developments and businesses in close proximity to the Antofagasta Port should be fully aware of the risk of contamination.

Houses and children living in close proximity to the Antofagasta Port should be constantly cleaned to avoid dust ingestion.

Long term: Stricter regulations on a national or regional level should be adopted to reduce dust-borne contamination. Also, businesses, schools, and prospective developments in close proximity to the port should develop relevant long-term strategies to minimize exposure to contaminated dust.

Polymetallic stockpiles and Cu concentrates should not be kept in the Antofagasta Port for long-term periods.

Investments should be made in medical training and infrastructure to properly remedy potential health impacts.

The implementation of a real-time monitoring program and high-quality geochemistry laboratory should be considered to improve the quantification of contaminant elements and their speciation in Antofagasta dust.

Supplemental Information

Supplemental Information 1 Tables as shown in the manuscript.

Tables related to the Antofagasta dust, including: Dust sites, Methodologies, Statistics, Geoaccumulation index, Enrichment factor, Hazard index, Bioaccesibility and City dust.

Click here for additional data file.

Supplemental Information 2 Raw data.

This table contains data of Antofagasta dust, rocks and weathering products of rocks of Antofagasta, and bioaccesibility of Antofagasta dust.

Click here for additional data file.

This study would have not been possible without help from the following: Verónica Oliveros and collaborators for facilitating the access to unpublished data (from Oliveros et al., 2007), Mario Pereira Acevedo and Manuel Arenas of the Servicio Nacional de Geología y Minería for providing 1:50,000 Antofagasta maps (National Service of Geology and Mining Service (SERNAGEOMIN), 2014), the Instituto de Salud Pública and Colegio Médico for providing dust data (ISP, 2014; Tchernitchin & Bolados, 2016), Brandon Schneider for English improvement, and Isel Cortés and CENMA for laboratory quality control information. We also thank David Massey and two anonymous reviewers for their valuable comments and suggestions.

Additional Information and Declarations

Competing Interests

Author Contributions

Data Availability

The authors declare that they have no competing interests.

Joseline S. Tapia conceived and designed the experiments, analyzed the data, contributed reagents/materials/analysis tools, prepared figures and/or tables, authored or reviewed drafts of the paper, approved the final draft.

Jorge Valdés analyzed the data, authored or reviewed drafts of the paper, approved the final draft.

Rodrigo Orrego analyzed the data, authored or reviewed drafts of the paper, approved the final draft.

Andrei Tchernitchin conceived and designed the experiments, performed the experiments, analyzed the data, contributed reagents/materials/analysis tools, prepared figures and/or tables, authored or reviewed drafts of the paper, approved the final draft, sampling.

Cristina Dorador analyzed the data, authored or reviewed drafts of the paper, approved the final draft.

Aliro Bolados conceived and designed the experiments, performed the experiments, analyzed the data, contributed reagents/materials/analysis tools, authored or reviewed drafts of the paper, approved the final draft, sampling.

Chris Harrod conceived and designed the experiments, analyzed the data, contributed reagents/materials/analysis tools, prepared figures and/or tables, authored or reviewed drafts of the paper, approved the final draft.

The following information was supplied regarding data availability:

Tapia, Joseline; Valdés, Jorge; Orrego, Rodrigo; Tchernitchin, Andrei; Dorador, Cristina; Bolados, Aliro; Harrod, Chris (2018): Raw_data.ods. figshare. Dataset. https://doi.org/10.6084/m9.figshare.5844366.v1.

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
