# Peer review of "Geologic and anthropogenic sources of contamination in settled dust of a historic mining port city in northern Chile: health risk implications"

_PeerJ, doi:10.7717/peerj.4699_

## Round 0.1 · original submission · Minor Revisions

I apologize for the the length of time this manuscript was in review. Please address all of the reviewer comments, particularly clarifications on analytical and instrumental procedures.

·

Basic reporting

After going trough the review of the manuscript I find it suitable or publication in this prestigious journal.

I think one reference can be added to the manuscript on page 16, line 216-217:

Massey, D., Kulshrestha, A., & Taneja, A., (2013). Particulate matter concentrations and their related metal toxicity in rural residential environment of semi-arid region of India. Atmospheric Environment, 67, 278-286.

Experimental design

'NO COMMENTS'

Validity of the findings

'NO COMMENTS'

Reviewer 2 ·

Basic reporting

The manuscript entitled " Geologic and anthropogenic sources of contamination in the settled dust of a historic mining port city in northern Chile: Health risk implications" evaluated the concentrations and health risk of 12 metals in street dust samples of the port of Antofagasta, Chile. The data from previous studies and the current one are well combined in this paper and useful results are obtained eventually. However, it is not very well organized and needs a few corrections. This paper includes original data on this environmentally polluted port city. The English language is good. Certainly, there are some questions and comments that need to be considered. Overall, I recommend a minor revision for reconsideration.
The following comments should be taken into consideration by the authors:
Comment 1# The authors present the Tables and figures of result section in “materials and methods” section, which is not common and this makes finding texts related to presented data difficult. Please relocate tables and figures. Moreover, it is necessary to number the titles and take some sections as sub-section of the others. For example, Antofagasta Region background, previous sampling campaigns, and Study objectives should be sub-titles of Introduction section.
Comment 2# Lines 348-354: This information is more related to introduction or materials and methods section. Please move them to those sections.

Comment 3# Line 372-428: The section “Recommendations and considerations” is too long and including the unnecessary explanation for international readers. The authors also mentioned the summary of this section at the end of conclusion part that is enough for international readers. So, I highly recommend removing this section.
Comment 4# Figure 4: in legend please correct the second range as 0.5<AP<1 km
Comment 5# Please change “Work Methodology” title to “Methodology”.

Experimental design

Comment 1: Line 225: Please mention which size fraction was used in bioaccessibility tests and add QC data. For example, what Standard reference material/materials were used? What are results for them? How many replicates were used for each sample?
Comment 2: Please justify using Fe as the reference element and add it to line 192.

Validity of the findings

no comment

Reviewer 3 ·

Basic reporting

The data is reported properly in the manuscript

Experimental design

Experimental design explained well in the manuscript except analytical and instrumental methods.

Validity of the findings

Findings are valid and presented well.

Additional comments

Current manuscript reports heavy metals levels and sources in street dust from Antofagasta region of Chile, where intensive mining activities exist. Current study reports an important data set in terms of assessment of human exposure to heavy metals. The manuscript is written well and data presented properly. I recommend this manuscript to be published after correction of the following issues

Line 110-119. Please state the size fraction of dust samples that wer subjected to analysis.
Line 140-142. Authors consider atmospheric deposition as the sole pathway for deposition of metals and metalloid in street dust. How about heavy metals from vehicle emissions? In rest of the manuscript, vehicles emissions were never mentioned as a source, especially for Pb. Since the manuscript is dealing with heavy metals in street dust, contribution of vehicle emissions should seriously considered.

General comment

There is no information on analytical and instrumental procedures, i.e. sample preparation, instrumental methods, quality assurance/quality control, detection limits etc. Such information should be given briefly in the manuscript.

---

## Round 0.2 · accepted · Accept

Thank you for your efforts in revising the manuscript.